# Obstructive Jaundice Induced by Hilar Mucinous Cystic Neoplasm of the Liver: A Rare Case Report and Literature Review

**DOI:** 10.3390/curroncol32030126

**Published:** 2025-02-23

**Authors:** Pengcheng Wei, Shengmin Zheng, Chen Lo, Yongjing Luo, Liyi Qiao, Jie Gao, Jiye Zhu, Yi Wang, Zhao Li

**Affiliations:** 1Department of Hepatobiliary Surgery, Peking University People’s Hospital, Beijing 100044, China; bjmuwpc@163.com (P.W.); pkuphzsm@126.com (S.Z.); chen-lo@outlook.com (C.L.); 1910301246@pku.edu.cn (Y.L.); qiaoliyi@sina.cn (L.Q.); gaojie_1131@163.com (J.G.); gandanwk@vip.sina.com (J.Z.); 2Beijing Key Surgical Basic Research Laboratory of Liver Cirrhosis and Liver Cancer, Beijing 100044, China; 3Peking University Center of Liver Cancer Diagnosis and Treatment, Beijing 100044, China; 4Peking University Institute of Organ Transplantation, Beijing 100044, China; 5Department of Hepatobiliary Surgery, Qingyang People’s Hospital, Qingyang 745099, China

**Keywords:** mucinous cystic neoplasm, benign liver tumor, hilar tumor, obstructive jaundice, diagnosis, surgical resection

## Abstract

Mucinous cystic neoplasm of the liver (MCN-L) is a rare benign tumor accounting for less than 5% of all liver cysts, with MCN-L in the hilar region being exceptionally uncommon and often misdiagnosed due to its complex presentation. A 48-year-old woman presented with obstructive jaundice following initial laparoscopic drainage of hepatic cysts, where pathology initially indicated benign cystic lesions. Months later, imaging revealed an enlarged cystic lesion in the left liver lobe with intrahepatic bile duct dilation. Further evaluations, including ultrasound, enhanced CT, and MRI, confirmed a large cystic lesion compressing the intrahepatic bile ducts. After a multidisciplinary discussion, hepatic cyst puncture and drainage were performed, temporarily alleviating jaundice. However, she returned with yellowish-brown drainage fluid and worsening jaundice, prompting cyst wall resection. Postoperative pathology confirmed MCN-L. Three months later, jaundice subsided, and a hepatic resection of segment 4 was performed, with pathology confirming low-grade MCN-L. At a 12-month follow-up, the patient showed no abnormalities. This case highlights the diagnostic and therapeutic challenges of MCN-L in the hilar region, as it can easily be mistaken for other liver cystic lesions on imaging. Pathologic examination is essential for definitive diagnosis, and early radical surgical resection is critical to improve prognosis and reduce the risk of malignancy and recurrence.

## 1. Introduction

Mucinous cystic neoplasm of the liver (MCN-L) is a rare benign tumor primarily originating in the intrahepatic biliary system, commonly affecting middle-aged women and accounting for less than 5% of all liver cysts [1,2,3]. Despite its low incidence, MCN-L poses significant diagnostic and therapeutic challenges, with a 3% to 5% chance of malignant transformation into cystadenocarcinoma [4]. The exact origin of MCN-L remains unclear, but it may be associated with ectopic ovarian-like stroma in the liver [3,4,5].

The literature on MCN-L is sparse, primarily consisting of case reports or small-sample studies. MCN-L located in the hilar region is even rarer and prone to bile duct obstruction, leading to obstructive jaundice. This condition severely impacts the patient’s quality of life and may cause various complications [6,7]. The clinical manifestations, imaging features, and pathological diagnosis of MCN-L are complex and heterogeneous, complicating clinical management and increasing the likelihood of misdiagnosis.

In this article, we report a rare case of obstructive jaundice caused by MCN-L in the hilar region. Through a review of the existing literature, we aim to summarize the clinical features, differential diagnosis, and therapeutic strategies of this disease to provide a valuable reference for clinicians.

## 2. Case Presentation

A 48-year-old female underwent a routine physical examination at a local hospital in October 2022, which revealed a hepatic cystic lesion. Imaging suggested a high likelihood of a simple hepatic cyst, and she subsequently underwent laparoscopic fenestration and drainage. Postoperative pathology confirmed a benign cystic lesion, and she was advised to undergo regular follow-up. In June 2022, follow-up abdominal CT showed an enlarged cystic lesion in the left hepatic lobe with intrahepatic bile duct dilation. However, as she was asymptomatic, no further intervention was undertaken. She remained on routine follow-up until September 2023, when she developed abdominal distension, generalized jaundice, and dark urine but without pruritus, clay-colored stools, and chills or a fever. She was then re-evaluated at the local hospital. Laboratory tests revealed alanine aminotransferase (ALT) 109 U/L, aspartate aminotransferase (AST) 78 U/L, total bilirubin (TBIL) 136.7 μmol/L, direct bilirubin (DBIL) 99.6 μmol/L, and carbohydrate antigen 19-9 (CA 19-9) 96.9 U/mL. Due to case complexity, the local hospital referred her to a tertiary center for further evaluation and treatment. Her medical history included two ces arean sections, total thyroidectomy for papillary thyroid carcinoma, and cholecystectomy, with no other significant conditions.

The patient was admitted to our hospital on 18 September 2023. Abdominal ultrasonography identified a large cystic lesion in liver segments S4/5, measuring approximately 11.2 × 9.6 cm, exhibiting anechoic characteristics with poor acoustic transmission. No significant color Doppler flow signals were detected. Intrahepatic bile duct dilation was noted in the left liver and part of the right liver. Contrast-enhanced CT and MRI revealed a large intrahepatic cystic mass (10.8 × 9.5 cm) with well-defined margins and no enhancement on contrast imaging. The lesion had mildly lobulated contours and significantly compressed the confluence of the left and right hepatic ducts, resulting in intrahepatic bile duct dilation, as shown in Figure 1. Due to the complexity of the recurrent hepatic cyst and its association with obstructive jaundice, a multidisciplinary team (MDT) discussion was held on 25 September 2023. The lesion was suspected to be a recurrent hepatic cyst, and fenestration with drainage was not recommended at this stage. Instead, ultrasound-guided cyst aspiration and drainage were planned to determine the cause of jaundice. If jaundice resolved or improved, drainage would be considered effective. However, if jaundice persisted or worsened, surgical exploration would be considered. On 26 September 2023, the patient underwent ultrasound-guided cyst aspiration and drainage, which yielded clear, colorless fluid. The patient experienced relief from abdominal distension and was discharged with a drainage catheter in place, instructed to undergo regular follow-up monitoring.

On 12 October 2023, the patient observed yellow-brown drainage fluid with an output of approximately 500 mL per day, leading to readmission for suspected biliary leakage. Laboratory tests revealed ALT 36 U/L, AST 44 U/L, gamma-glutamyl transferase (GGT) 52 U/L, alkaline phosphatase (ALP) 122 U/L, TBIL 213.4 μmol/L, and DBIL 160.8 μmol/L, confirming worsening obstructive jaundice. On 16 October 2023, an MDT discussion concluded that persistent bile duct compression by the hepatic cyst and newly developed biliary leakage after cyst aspiration required surgical intervention. On 18 October 2023, the patient underwent exploratory surgery under general anesthesia. Intraoperatively, a large cystic lesion was found in liver segments S4/5, with part of the cyst wall extending into the gallbladder bed. Some parts of the cyst wall were thickened and covered by liver tissue. A localized bile collection was noted between the right hepatic lobe and diaphragm, with a drainage tube positioned inside. Abdominal adhesions were observed. A puncture site was found on the diaphragmatic surface of segment S5, with evident bile leakage. The common bile duct had a normal diameter, but the hepatic cyst compressed the common hepatic duct dorsally. The procedure involved adhesiolysis, aspiration of the subphrenic bile collection, and irrigation with normal saline. The cyst wall was excised from the liver using electrocautery, revealing clear cystic fluid and a smooth inner surface. The remaining cyst wall margins were sutured or ligated. As the gallbladder bed was part of the cyst wall, the cystic artery and cystic duct were ligated and divided during resection. The common bile duct was located at the hepatoduodenal ligament, and a longitudinal incision was made on its anterior wall for choledochoscopy. The common hepatic duct, left and right hepatic ducts, and second- and third-order branches were free of neoplastic lesions, strictures, or stones. The distal common bile duct and Oddi’s sphincter were patent. A T-tube was placed in the common bile duct. The incision was sutured and tested for leakage, confirming no bile leakage from the residual cyst wall. Postoperative pathology confirmed low-grade MCN-L, with a cyst wall measuring 6.5 × 3.5 × 1 cm. Immunohistochemistry showed CK (+), CK7 (+), CK19 (+), CK20 (−), ER (+), PR (+), and PAX8 (−).Due to the risk of malignant transformation, the patient was advised to undergo a second-stage surgery after three months. The abdominal drainage tubes near the common bile duct and right hepatic region were removed on postoperative days 7 and 13. The T-tube was removed six weeks postoperatively, with no reported discomfort.

On 20 January 2024, the patient was readmitted for follow-up evaluation and further treatment. Laboratory tests revealed ALT 68 U/L, AST 44 U/L, GGT 29 U/L, ALP 58 U/L, TBIL 10.8 μmol/L, and DBIL 3.7 μmol/L, confirming complete jaundice resolution. Due to the potential risk of malignant transformation in MCN-L, we conducted a detailed discussion with the patient and opted for curative resection to minimize recurrence and malignant progression. On 24 January 2024, the patient underwent a repeat surgery under general anesthesia. Intraoperatively, adhesions between the liver and diaphragm were meticulously dissected. A 5 cm cystic lesion was identified in segment 4 of the liver, with well-defined margins. Segment 4 hepatectomy was performed, with meticulous removal of the cyst wall tissue covering the left and right hepatic ducts. The residual cyst wall was ablated using an argon plasma coagulator, and a single abdominal drainage tube was placed at the liver transection site. Postoperative pathology confirmed low-grade MCN-L, with mucinous columnar epithelial lining, hemorrhage, cholesterol crystal deposition, and multinucleated foreign body giant cells. Ovarian-like stroma was present, and mild steatosis was observed in the surrounding liver parenchyma. The patient was discharged on postoperative day 5, and the abdominal drainage tube was removed after two weeks. At the 12-month follow-up, no recurrence was detected, as shown in Figure 2 and Figure 3.

## 3. Discussion

MCN-L is a rare benign tumor first documented by Edmondson in 1958, with few cases reported in the literature. Wheeler et al. in 1985 observed that cystic and mucus-producing tumors in the liver histopathologically featured an ovarian-like stroma [8]. The 2002 WHO Classification of Tumors of the Digestive System referred to these as biliary cystadenomas and cystadenocarcinomas, but this was deemed inappropriate due to the characteristic ovarian-like mesenchyme. Clinical symptoms, serologic markers, and imaging often fail to reliably diagnose these tumors, leading to frequent misdiagnosis [9,10]. In 2010, WHO reclassified these tumors, and by the 2019 edition, they were recognized as MCN of the liver and biliary tract [11,12]. Despite these clarifications, the rarity of MCN-L continues to pose challenges for early and accurate diagnosis. Predominantly affecting middle-aged females [2,13], MCN-L typically presents with abdominal discomfort and may cause elevated serum CA19-9 levels, leading to complications such as hepatic impairment and bile duct compression [6,14]. Our patient, a middle-aged female, presented with elevated transaminase and CA19-9 levels, as well as obstructive jaundice due to a tumor in the hilar region, consistent with previous reports.

Preoperative diagnosis of MCN-L is challenging due to its similarities in imaging and clinical presentation with other cystic liver lesions, such as simple cysts and choledochal cystadenocarcinomas, leading to a high misdiagnosis rate [15]. The complexity of diseases causing obstructive jaundice, combined with the rarity of MCN-L-induced obstructive jaundice, exacerbates diagnostic and treatment difficulties, as illustrated in Table 1. In this case, the patient was initially misdiagnosed with a simple hepatic cyst at a local hospital and underwent laparoscopic fenestration and drainage. The tumor’s location in the hilar region and postoperative pathology indicating only a benign cystic lesion delayed further treatment, resulting in recurrent symptoms. Upon presenting to our hospital with obstructive jaundice, the cystic mass was initially considered a recurrent hepatic cyst based on past medical history. However, postoperative pathology revealed low-grade MCN-L, which was difficult to diagnose preoperatively without detailed pathological examination. A review of the initial pathology specimen from the local hospital confirmed MCN-L, highlighting the diagnostic challenges at less specialized facilities. This case underscores the complexity of diagnosing MCN-L and the limitations of serologic markers and imaging tests. Elevated CA19-9 levels, while indicative of MCN-L, have low specificity and can be confused with other hepatobiliary diseases. Imaging modalities such as ultrasound, CT, and MRI provide important clues but are limited in assessing the cystic wall and internal structure. Therefore, imaging findings must be comprehensively analyzed alongside clinical manifestations and pathological features [9,16].

There are relatively few descriptions and studies of MCN-L in both literature reviews and clinical work, but each case report provides clinicians with valuable experience and references. The diagnosis of MCN-L requires differentiation from various cystic liver lesions, as illustrated in Table 2, as follows: (1) Endometriotic cysts: Patients often have a history of endometriosis or pelvic surgery. Imaging shows single or multicompartmental cystic masses. Pathology shows endometrial epithelium and mesenchyme, positive CD10, ER, PR, and no communication with bile ducts (2) Hepatic encapsulated cysts: Often seen in patients with a history of exposure to infected areas. Larger lesions can cause abdominal distension, vague pain, and bile duct compression, leading to obstructive jaundice. Imaging features include a thick cyst wall with calcification or a “cyst within a cyst” multilocular pattern and no enhancement. Pathology shows a fibrous envelope and parasite body. (3) Simple hepatic cysts: Mostly found in women and are typically asymptomatic. Larger cysts can cause loss of appetite, nausea, and vomiting. When located in the porta hepatis, they can compress bile ducts and cause jaundice. Imaging features include round or ovoid low-density shadows with thin walls, clear boundaries, and no enhancement. Pathology shows single-layer cuboidal or columnar epithelium and clear cystic fluid. (4) Intraductal papillary neoplasm of bile duct (IPNB): Most common in males, often presenting with abdominal pain, distension, fever, and jaundice. Imaging shows cystic dilatation of bile ducts and nodules in the lumen, with the enhancement scan showing a “fast in, fast out” sign. Pathology shows bile duct epithelial hyperplasia, lack of ovarian-like mesenchyme, and negative ER, PR. (5) Bacterial liver abscess: Patients present with chills, high fever, pain in the liver area, malaise, loss of appetite, nausea, and vomiting. Imaging shows low-density foci, abscess wall density higher than the cavity but lower than liver tissue, and ring enhancement of the abscess wall. Pathology shows liver tissue destruction, many leukocytes, and cellular debris. (6) Hepatoportal cholangiocarcinoma: Most common in middle-aged and elderly men aged 50–70 years, presenting with progressive jaundice, itchy skin, weight loss, and abdominal pain. Imaging shows irregular thickening of the bile duct wall, dilatation of intrahepatic bile ducts, and delayed enhancement. Pathology shows bile duct adenocarcinoma, with CK7 and CK20 positive in immunohistochemistry. (7) Mucinous cystic adenocarcinoma: Most common in older men. Imaging features include a soft tissue shadow of the cyst wall and enhancement. MCN-L has the potential for malignant transformation into cystic adenocarcinoma. These differential diagnoses ensure comprehensive consideration of MCN-L and reduce the risk of misdiagnosis, ensuring accurate diagnosis and appropriate treatment.

Pathological examination is crucial for confirming the diagnosis of MCN-L, an epithelial neoplasm characterized by cyst formation. The cyst’s inner wall is lined with cuboidal or columnar mucinous epithelial cells, underlain by ovarian-like mesenchymal stroma containing ER and PR [4,11,36,37]. MCN-L is classified into low-grade/moderate-grade heterogeneous hyperplasia, high-grade heterogeneous hyperplasia, or invasive carcinoma based on epithelial cell heterogeneity. Our patient’s postoperative pathology revealed low-grade MCN-L with ovarian-like mesenchyme, confirming the diagnosis. Few studies exist on the histopathology and cytology of MCN-L. A multicenter study by Van et al. indicated that MCN-L mesenchyme exhibits features of ovarian mesenchyme with hormone responsiveness and estrogen synthesis at morphological, RNA, and protein levels. The Hedgehog and Wnt signaling pathways were significantly upregulated, suggesting that these tumors may originate from cells with a transdifferentiation ability. These findings highlight ovarian-like mesenchyme as a crucial differential diagnostic marker for MCN-L and provide new insights into its biology and pathogenesis [4]. However, while histopathology and cytology aid in tumor diagnosis, tumor puncture should be approached with caution due to potential risks. The pros and cons of performing tumor puncture prior to surgery must be carefully considered [16].

For patients with MCN-L, cyst drainage may be used for diagnostic or palliative purposes in select cases; however, its clinical application remains controversial. In this case, the patient initially presented with a suspected hepatic cyst recurrence. After an MDT discussion, ultrasound-guided cyst aspiration and drainage were performed, temporarily resolving jaundice. However, cyst drainage has several limitations: (1) it is not a definitive treatment for MCN-L, and recurrence or progression may occur; (2) cyst aspiration can cause cyst content spillage into the bile duct, increasing the risk of biliary infection or obstruction; (3) it may lead to tumor cell dissemination, especially in lesions with malignant potential. Therefore, cyst drainage is generally not recommended as a first-line treatment and should only be used for preoperative cyst fluid analysis or in patients unfit for surgery. In this case, although jaundice initially subsided after aspiration, the drainage fluid later turned yellowish-brown, and the patient developed recurrent obstructive jaundice, suggesting that drainage alone was inadequate to relieve bile duct compression, requiring further surgical intervention. Cyst fluid biochemical analysis can provide valuable insights in preoperative diagnosis. Studies indicate that elevated CA19-9 and CEA levels in cyst fluid can help distinguish neoplastic from non-neoplastic lesions [15,38,39]. In MCN-L, cyst fluid CA19-9 and CEA levels are typically markedly elevated, whereas in simple hepatic cysts or infectious cystic lesions, these tumor markers remain low. Therefore, cyst fluid analysis is a valuable tool for preoperative differential diagnosis. However, the limited specificity of CA19-9 and CEA means their elevation may also be observed in cholangiocarcinoma or inflammatory conditions. Thus, a comprehensive assessment combining imaging, pathology, and clinical presentation is essential for accurate diagnosis.

Cases of MCN-L causing obstructive jaundice are rare and demonstrate significant heterogeneity in presentation. The present case, along with those reported by Siriwardana et al. [22] and Li et al. [6], involved tumor-induced biliary obstruction but differed in anatomical location, clinical presentation, and therapeutic strategies. Siriwardana et al. described a 25-year-old female with a segment 4 liver tumor, where part of the cyst wall protruded into the left hepatic duct, causing episodic biliary obstruction with intermittent jaundice and pruritus. This contrasts with the persistent obstructive jaundice observed in the present case. The patient underwent left hepatectomy and remained recurrence-free during follow-up. In contrast, Li et al. reported a 56-year-old female with a cystic tumor affecting the left liver and hilar bile duct, with tumor fragments dislodging into the common bile duct. MRI showed bile duct wall thickening, and CA19-9 levels were elevated. She underwent left hepatectomy with bile duct resection and choledochojejunostomy. The present case is unique as the MCN-L was primarily located in the hilar region. The initial surgery involved only partial cyst wall resection. However, due to the tumor’s proximity to the bile duct, residual tissue remained postoperatively, necessitating a subsequent segment 4 hepatectomy. Compared to the present case, Siriwardana et al.’s case did not involve direct bile duct invasion, allowing for single-stage curative resection. In contrast, Li et al.’s case required extensive bile duct resection and biliary reconstruction due to common bile duct involvement. This case highlights the complexity of MCN-L in the hilar region, particularly in surgical decision making, where balancing limited resection with complete tumor removal is crucial to minimizing recurrence and malignant transformation risk.

Radical surgical resection is the treatment of choice for MCN-L, often resulting in a favorable prognosis [40]. Because MCN-L carries a potential risk of malignancy, early surgical resection reduces this risk and improves the chances of being cured. Even if the tumor has progressed to malignancy, the 5-year survival rate after radical resection can exceed 70% [41,42]. Complete resection of cystic liver masses is recommended when preoperative diagnosis is unclear but MCN-L is suspected, as simple liver cysts risk recurrence with only open drainage. In our case, the patient underwent laparoscopic fenestration and drainage at a local hospital due to an unclear diagnosis, leading to recurrent symptoms and multiple subsequent surgeries. Fortunately, radical surgical resection resulted in a favorable prognosis, with no significant abnormalities observed 12 months postoperatively. Early radical resection reduces the risk of malignancy, recurrence, and progression, while providing a definitive diagnosis and detailed pathologic evaluation, guiding prognosis more precisely. A retrospective study by Shi et al. showed that one patient with MCN-L developed recurrence and metastasis 9 months after surgery due to invasive cancer, demonstrating a high risk of recurrence and metastasis in invasive MCN-L. The other patients survived during the follow-up period, with a maximum survival of up to 60 months [16]. Therefore, close postoperative follow-up and surveillance are crucial for timely detection and management of recurrence and metastasis.

For patients who cannot tolerate surgery or are unsuitable for extensive resection, minimally invasive treatment strategies may serve as alternative or transitional options. In recent years, therapeutic endoscopic ultrasound (EUS) has gained prominence in managing biliary obstruction. Specifically, for patients with failed endoscopic retrograde cholangiopancreatography (ERCP) or complex anatomy, EUS-guided biliary drainage (EUS-BD) has emerged as a minimally invasive alternative [43]. Although EUS-BD has shown high technical success and low postoperative complication rates in malignant biliary obstruction, its role in benign cases requires further investigation. For nonsurgical candidates, percutaneous transhepatic biliary drainage (PTBD) can provide temporary relief but may cause infection or drainage dependence with prolonged use. Other minimally invasive approaches, including biliary stent placement and percutaneous cyst aspiration with sclerotherapy, may serve as palliative or transitional treatments, though their long-term efficacy remains uncertain. Liver transplantation may be an option for patients with end-stage liver disease or extensive bile duct involvement; however, evidence supporting its use for localized MCN-L remains insufficient. This case underscores the importance of individualized treatment for hilar MCN-L, prioritizing complete tumor resection while minimizing unnecessary invasive procedures. Advances in minimally invasive techniques may further enhance patient outcomes.

In summary, our case presents a typical instance of obstructive jaundice caused by MCN-L in the hilar region. The patient underwent three surgeries, through which we gained valuable experience in diagnosing and treating MCN-L. To improve the diagnostic rate of MCN-L, regional hospitals should strengthen the use of pathological and imaging examinations combined with MDT discussions to enhance diagnostic accuracy. With advancements in molecular biology and genetics, more biomarkers are expected to be used for prognostic assessment and treatment selection in the future. We are optimistic about the future development of this field and anticipate further research results and clinical applications. Future research should address current limitations by incorporating larger sample sizes, adopting multicenter designs, and integrating molecular biology approaches. These improvements will enable a more comprehensive exploration of diagnostic and therapeutic strategies for MCN-L, thereby improving patient prognosis and quality of life.

## 4. Conclusions

This case of MCN-L in the hilar region causing obstructive jaundice highlights its diagnostic and therapeutic challenges. As MCN-L resembles other liver cystic lesions in imaging and clinical manifestations, it is easily misdiagnosed. Pathologic examination plays a crucial role in confirming the diagnosis. Early radical surgical resection is essential to improve prognosis and reduce the risk of malignancy and recurrence.

## Figures and Tables

**Figure 1 curroncol-32-00126-f001:**
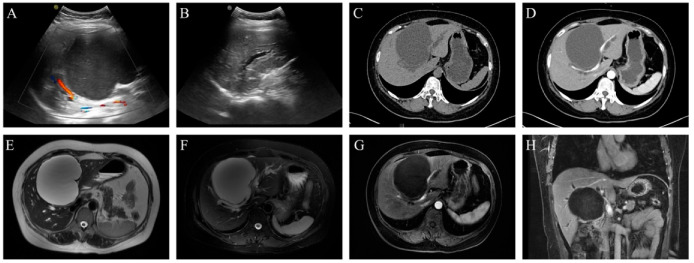
Imaging findings of the patient. (**A**,**B**) Abdominal ultrasound shows a large, anechoic area in the liver (primarily in segments S4/5), measuring approximately 11.2 × 9.6 cm, with poor internal sound transmission and no detectable color flow signals. The left and part of the right intrahepatic bile ducts are dilated, with a maximum width of approximately 0.5 cm. (**C**,**D**) Contrast-enhanced CT reveals a large cystic density lesion in the liver, measuring approximately 10.8 × 9.5 cm. The unenhanced CT value is about 10 HU, and no enhancement is observed after contrast administration. (**E**–**H**) Contrast-enhanced MRI demonstrates a large cystic lesion with a water-like signal in the liver, measuring approximately 10.8 × 9.5 cm, with clear boundaries and slightly lobulated edges. The lesion compresses the confluence of the left and right intrahepatic bile ducts, causing significant intrahepatic bile duct dilatation, while the extrahepatic bile ducts remain unaffected.

**Figure 2 curroncol-32-00126-f002:**
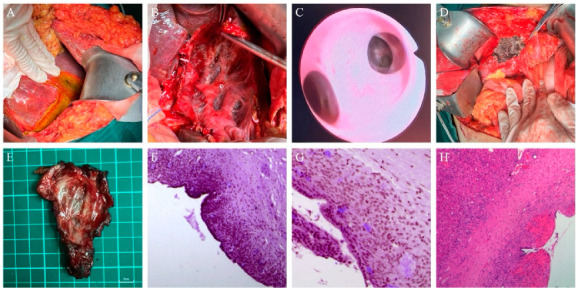
Surgical procedures and postoperative pathological findings. (**A**,**B**) In the initial surgery, a large cystic lesion is observed in liver segments S4/5. The cyst wall is dissected along its margin with the liver. (**C**) Intraoperative cholangioscopy during the initial surgery to rule out the possibility of intraductal lesions. (**D**) In the second surgery, segment 4 of the liver is resected, and the residual surface is ablated using an argon plasma coagulator. (**E**) Gross specimen from the second surgery showing the resected liver segment. (**F**) HE staining of the cyst wall resection specimen from the initial surgery (magnification ×40). (**G**) HE staining of the cyst wall resection specimen from the initial surgery (magnification ×100), showing tissue edema and focal lymphocytic infiltration. (**H**) HE staining of the liver resection specimen from the second surgery (magnification ×40), revealing fibrous cystic wall-like tissue partially lined by mucinous columnar epithelium. Hemorrhage, cholesterol crystal deposition, and multinucleated foreign body giant cells are observed, consistent with a low-grade mucinous cystic neoplasm. The surrounding liver parenchyma exhibits mild steatosis.

**Figure 3 curroncol-32-00126-f003:**
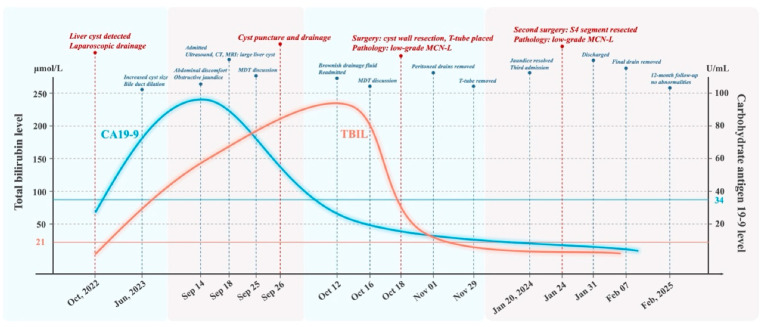
Timeline of clinical events and total bilirubin (TBIL) and carbohydrate antigen 19-9 (CA19-9) levels. The normal upper reference limits are 21 µmol/L for TBIL and 34 U/mL for CA19-9.

**Table 1 curroncol-32-00126-t001:** Reported Cases of Mucinous Cystic Neoplasm of the Liver (MCN-L) Causing Obstructive Jaundice, Including the Present Case.

No.	Age	Gender	TBIL (umol/L)	CA 19-9 (IU/mL)	Tumor Localization	Size (cm)	Symptom	Treatment	Follow-Up	Prognosis	Author	Year
1	58	F	283	1188	Left lobe	NA	Jaundice, abdominal pain	Left lobectomy	12 months	Alive without recurrence	Preetha et al. [17]	2004
2	40	F	N/A	N/A	Left lobe	7	Jaundice, abdominal pain	Left lobectomy	6 months	Alive without recurrence	Baudin et al. [18]	2006
3	43	F	18.8	51.6	Left lobe (S2/S3)	10	Jaundice, abdominal pain	Segment II/III resection, cholecystectomy	N/A	N/A	Seidel et al. [19]	2007
4	58	M	133.4	107	Left lobe	NA	Jaundice, abdominal pain	Left lobectomy, cholecystectomy	N/A	N/A	Yu et al. [20]	2008
5	56	F	434	92.1	Left lobe (S4)	5.5	Jaundice, abdominal pain	Left lobectomy	12 months	Alive without recurrence	Yi et al. [21]	2009
6	25	F	N/A	Normal	Left lobe (S4b)	5.5	Jaundice, abdominal pain, pruritus	Left lobectomy, cholecystectomy	24 months	Alive without recurrence	Siriwardana et al. [22]	2009
7	32	F	N/A	43	Left lobe (S3)	7.9	Jaundice, abdominal pain, diarrhea	Left lobectomy	N/A	N/A	Gonzalez et al. [23]	2009
8	40	F	50	N/A	Hilar	7	Jaundice, abdominal pain	Left lobectomy	12 months	Alive without recurrence	Jawad et al. [24]	2009
9	57	F	N/A	>7000	Left lobe (S4)	5	Jaundice, abdominal pain	Left lobectomy, cholecystectomy	12 months	Alive without recurrence	Harmouch et al. [25]	2011
10	41	F	N/A	N/A	Left lobe (S3/S4)	5	Jaundice, abdominal pain	Left lobectomy, bile duct resection	18 months	Alive without recurrence	Vyas et al. [26]	2011
11	53	F	193.2	Normal	S4/S8	10	Jaundice	Liver transplantation	48 months	Alive without recurrence	Romagnoli et al. [27]	2011
12	29	F	310	>29,000	Left lobe	8	Jaundice, abdominal pain	Left lobectomy	12 months	Alive without recurrence	Cecka et al. [28]	2011
13	62	F	19	Normal	Left lobe	NA	Jaundice, dysuria	Left lobectomy, bile duct resection	24 months	Alive without recurrence	Soochan et al. [29]	2012
14	28	F	N/A	N/A	Left lobe (S4)	7.3	Jaundice, abdominal pain	Left lobectomy, bile duct resection, cholecystectomy	N/A	N/A	Abe et al. [30]	2012
15	37	F	54.7	Normal	Left lobe	2.9	Jaundice, abdominal bloating	Left lobectomy, bile duct resection	N/A	N/A	Rayapudi et al. [31]	2013
16	39	F	N/A	>1000	Left lobe (S4)	NA	Jaundice, mild hepatomegaly, pruritus	Left lobectomy	3 months	Alive without recurrence	Chandrasinghe et al. [7]	2013
17	57	F	46.2	99	Left lobe (S4)	8.3	Jaundice, abdominal pain, fever	Left lobectomy, cholecystectomy	N/A	N/A	Takano et al. [32]	2015
18	26	F	70.1	Normal	Left lobe (S4)	6.1	Jaundice	Left lobectomy	N/A	N/A		
19	78	F	64.4	Normal	Left lobe	NA	Jaundice, fever	Left lobectomy	N/A	N/A	Li et al. [33]	2016
20	20	F	N/A	N/A	Left lobe (S4)	6	Jaundice, abdominal pain, pruritus, fever	Left lobectomy	2 months	Alive without recurrence	Anand et al. [34]	2019
21	28	F	N/A	N/A	Left lobe (S4)	5	Jaundice, pruritus, anorexia	Left lobectomy, bile duct resection	12 months	Alive without recurrence		
22	31	F	79	N/A	Left lobe (S4b)	4.4	Jaundice, abdominal pain, vomiting, fever	Left lobectomy, bile duct resection, cholecystectomy	6 months	Alive without recurrence	Srinivas et al. [35]	2020
23	56	F	149.1	189	Left lobe	7.4	Jaundice, abdominal pain, anorexia	Left lobectomy, bile duct resection, cholecystectomy	N/A	N/A	Li et al. [6]	2023
24	48	F	136.7	96.9	Hilar	11.2	Jaundice, abdominal pain	Segment IV resection	12 months	Alive without recurrence	Present Case	2024

N/A, not available; F, female; M, male; TBIL, total bilirubin; CA 19-9, carbohydrate antigen 19-9.

**Table 2 curroncol-32-00126-t002:** Differential Diagnosis of Mucinous Cystic Neoplasm of the Liver (MCN-L): Clinical and Pathological Features.

Disease Name	Main Symptoms	Laboratory Findings	Imaging Features	Pathological Features	Treatment Methods
MCN-L	Abdominal pain, bloating, loss of appetite, nausea, obstructive jaundice if bile duct compression occurs	Elevated CA19-9	Unilocular/multilocular low-density cystic lesion, thin wall, septation or wall calcification; enhanced septa on contrast scan; no bile duct dilation	Cyst wall lined with cuboidal/columnar tumor epithelium, focal mucin staining positive, varying epithelial atypia, ovarian-like stroma with ER/PR positive	Anatomic liver resection or hemihepatectomy
Endometriotic Cyst	Non-specific symptoms, history of endometriosis or pelvic surgery	Elevated CA125, CA19-9	Single/multilocular cystic mass	Endometrial epithelium and stroma, cluster of differentiation 10 (CD10), estrogen receptor (ER), progesterone receptor (PR) positive, not connected to bile ducts	Hormonal therapy or surgical removal
Hepatic Hydatid Cyst	Asymptomatic when small; bloating, dull pain, obstructive jaundice if large	Positive echinococcosis serology	Low-density lesion, thick wall with calcification or “cyst within a cyst”; no enhancement on contrast scan	Fibrous outer cyst and inner hydatid cyst	Surgery (primary), medication (adjunct); complete cyst removal or liver segment resection
Simple Hepatic Cyst	Mostly asymptomatic, large cysts may cause appetite loss, nausea, vomiting, obstructive jaundice if bile duct compression occurs	Unremarkable	Round/oval low-density shadow, extremely thin wall, clear boundaries, no enhancement on contrast scan	Single-layer cuboidal/columnar epithelium, clear cyst fluid, not connected to bile ducts	Cyst fenestration or aspiration and sclerosis
IPNB	Abdominal pain, bloating, fever, jaundice	Elevated CEA, CA19-9	Cystic bile duct dilation with intraluminal nodules, “fast-in, fast-out” enhancement	Biliary-type epithelial hyperplasia, no ovarian-like stroma, ER/PR negative, papillary structures, connected to bile ducts	Segmental/lobar liver resection, hepatectomy + extrahepatic bile duct resection if needed
Pyogenic Liver Abscess	Chills, high fever, right upper quadrant pain, fatigue, appetite loss, nausea, vomiting	Elevated WBC, infection markers, transaminases, bilirubin	Low-density lesion, abscess wall density higher than abscess cavity but lower than liver tissue, ring enhancement of abscess wall	Liver tissue destruction, many WBCs, cell debris, necrotic tissue, granulation and fibrosis in chronic cases, can be connected to bile ducts	Antibiotics, percutaneous drainage under ultrasound guidance, surgical drainage for severe cases
Perihilar Cholangiocarcinoma	Progressive jaundice, pruritus, weight loss, abdominal pain	Elevated bilirubin, liver enzymes, CA125, CA19-9, CEA	Irregular bile duct wall thickening, possible liver parenchyma involvement, significant bile duct dilation, delayed enhancement of lesion on contrast scan	Cholangiocarcinoma, cytokeratin 7 (CK7), cytokeratin 20 (CK20), mucin 1 (MUC-1) positive on immunohistochemistry	Radical resection
Mucinous Cystadenocarcinoma	Abdominal pain, jaundice	Elevated CA19-9	Soft tissue shadow in cyst wall with enhancement on contrast scan	Significant epithelial atypia, tumor cells may invade outside cyst wall	Radical resection

MCN-L, mucinous cystic neoplasm of the liver; IPNB, intraductal papillary neoplasm of the bile duct; CA19-9, carbohydrate antigen 19-9; CA125, cancer antigen 125; CEA, carcinoembryonic antigen; WBC, white blood cell count.

## Data Availability

No new data were created or analyzed in this study. Data sharing does not apply to this article.

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
