# Peer review of "Obstructive Jaundice Induced by Hilar Mucinous Cystic Neoplasm of the Liver: A Rare Case Report and Literature Review"

_curroncol, 2025, doi:10.3390/curroncol32030126_

Round 1
Reviewer 1 Report
Comments and Suggestions for Authors
Wei et al described an interesting case with a thorough literature review. I have some comments that need to be addressed.
There is a discrepancy between the case description and the discussion. The information about the local and reference centres and their management should be included in the case description. Furthermore, the reason for the patient's readmission on January 20th is not mentioned, even though liver function tests were normal.
The authors provide a comprehensive review of the differential diagnosis; however, they do not discuss the advantages and disadvantages of cyst drainage or consider whether the analysis of CA 19.9 and CEA in the intracystic fluid would be recommended within the diagnostic algorithm.
The authors should propose a diagnostic algorithm and provide an extensive discussion of therapeutic management, addressing both surgical and non-surgical candidates, as well as the role of liver transplantation in this context.
Author Response
Dear Reviewer,
We sincerely appreciate your thoughtful comments and constructive suggestions, which have significantly improved the quality of our manuscript. Below, we provide a point-by-point response to address each of your concerns.
- Discrepancy between the Case Description and the Discussion
Reviewer’s Comment: There is a discrepancy between the case description and the discussion. The information about the local and reference centres and their management should be included in the case description. Furthermore, the reason for the patient's readmission on January 20th is not mentioned, even though liver function tests were normal.
Response: We have revised the case description to include additional details regarding the patient's initial diagnosis and treatment at the local hospital. Specifically, we have clarified that the patient underwent surgical treatment at a local hospital, experienced tumor recurrence, and was subsequently referred to our hospital for further evaluation and management due to the complexity of the condition. Additionally, we have now explicitly stated the reason for the patient’s readmission on January 20th, emphasizing that, despite normal liver function tests, curative resection was performed after thorough discussion with the patient to minimize the risk of recurrence and malignant transformation associated with MCN-L.
- Discussion on the Advantages and Disadvantages of Cyst Drainage and the Role of CA19-9 and CEA Analysis
Reviewer’s Comment: The authors provide a comprehensive review of the differential diagnosis; however, they do not discuss the advantages and disadvantages of cyst drainage or consider whether the analysis of CA 19-9 and CEA in the intracystic fluid would be recommended within the diagnostic algorithm.
Response: In accordance with your suggestion, we have expanded the Discussion section to include a detailed analysis of the advantages and disadvantages of cyst drainage in MCN-L cases. We discuss its potential role in symptomatic relief and preoperative evaluation while highlighting the risks associated with recurrence, bile leakage, and possible tumor cell dissemination. Furthermore, we have incorporated a discussion on CA 19-9 and CEA levels in intracystic fluid, emphasizing their utility in distinguishing neoplastic from non-neoplastic cystic lesions, along with their limitations due to potential overlap with other cystic hepatic lesions. These revisions help to enhance the completeness and clinical applicability of our diagnostic approach.
- Diagnostic Algorithm and Comprehensive Discussion of Therapeutic Management
Reviewer’s Comment: The authors should propose a diagnostic algorithm and provide an extensive discussion of therapeutic management, addressing both surgical and non-surgical candidates, as well as the role of liver transplantation in this context.
Response: In response to this recommendation, we have included a comparative discussion of similar cases from the literature, allowing for a more comprehensive understanding of the diagnostic and therapeutic challenges associated with MCN-L. Additionally, we have expanded the discussion on various treatment modalities, including:
- Surgical resection, as the definitive treatment for MCN-L, with indications and considerations based on tumor size, location, and presence of biliary involvement.
- Minimally invasive approaches, such as cyst aspiration and drainage, which may provide temporary relief but are associated with a high recurrence risk and potential complications.
- Liver transplantation, which is rarely required but may be considered in cases with diffuse hepatic involvement or extensive biliary tract obstruction, though current evidence supporting this approach remains limited.
By integrating this comparative analysis and treatment discussion, we believe that the manuscript now offers a more comprehensive and structured approach to the management of MCN-L, benefiting both clinicians and researchers.
Final Remarks
Once again, we sincerely appreciate the reviewer’s valuable feedback. These revisions have significantly improved our manuscript, making it more comprehensive and clinically relevant. We hope that our responses adequately address your concerns, and we look forward to your further comments.

Reviewer 2 Report
Comments and Suggestions for Authors
This case report discusses a rare presentation of hilar mucinous cystic neoplasm (MCN) of the liver causing obstructive jaundice. The manuscript is comprehensive, detailing the clinical presentation, diagnostic challenges, and therapeutic management, supported by imaging and pathology findings. It provides valuable insights into the complexity of managing MCN-L, which aligns well with the journal's scope. However, the manuscript could be improved with some additional details and revisions to contextualize its findings better and incorporate recent advancements in the management of obstructive jaundice and the role of endoscopic techniques.
Major Comments
o The authors emphasize the rarity of hilar MCN-L and its diagnostic challenges, which is appropriate. However, to strengthen the manuscript, the authors should provide a more in-depth comparison of this case with other similar cases reported in the literature, particularly focusing on differences in management approaches.
o Given the increasing role of therapeutic endoscopic ultrasound (EUS) in managing biliary obstructions, the authors should address this in the discussion. Citing recent studies, such as PMID: 39768654 (on EUS-guided biliary drainage in malignant double obstruction), would enrich the discussion by providing an updated perspective on alternative management strategies for obstructive jaundice.
o The manuscript provides a good description of imaging and pathological findings. However, adding annotated figures for histological slides and imaging results (e.g., marking specific features like cystic wall thickening or ovarian-like stroma) would improve clarity for readers.
o The manuscript highlights the importance of early surgical resection in preventing malignancy and recurrence. It would benefit from a brief discussion on the risks and benefits of less invasive alternatives, such as percutaneous or endoscopic approaches, when surgery is not feasible.
Minor Comments
o The manuscript is mostly clear but contains minor grammatical errors. For example, “the MBO to be a serious threat” should be clarified.
o For smoother readability, sentences such as “the cyst wall and surface hepatic tissues were resected” could be rewritten.
o Ensure consistent use of terms, e.g., "MCN-L" instead of alternating between "MCN-L" and "hilar MCN."
o The timeline figure (Figure 3) is helpful but could be simplified for better readability. Consider enlarging key labels and focusing on the most critical clinical events.
Author Response
Dear Reviewer,
We sincerely appreciate your valuable comments and thoughtful suggestions, which have helped us improve the clarity and completeness of our manuscript. Below, we provide a point-by-point response addressing each of your concerns and detailing the corresponding revisions made in the manuscript.
Major Comments
- Comparison with Similar Cases in the Literature
Reviewer’s Comment: The authors emphasize the rarity of hilar MCN-L and its diagnostic challenges, which is appropriate. However, to strengthen the manuscript, the authors should provide a more in-depth comparison of this case with other similar cases reported in the literature, particularly focusing on differences in management approaches.
Response: As suggested, we have introduced a comparative analysis of our case with two similar cases reported by Siriwardana et al. and Li et al. This comparison highlights differences in clinical presentation, tumor location, and management strategies, thereby providing a more robust discussion of potential diagnostic and therapeutic approaches. This additional discussion has been structured into a dedicated paragraph in the Discussion section, ensuring logical coherence and stronger theoretical support.
- Role of Therapeutic Endoscopic Ultrasound (EUS) in Biliary Obstruction
Reviewer’s Comment: Given the increasing role of therapeutic endoscopic ultrasound (EUS) in managing biliary obstructions, the authors should address this in the discussion. Citing recent studies, such as PMID: 39768654 (on EUS-guided biliary drainage in malignant double obstruction), would enrich the discussion by providing an updated perspective on alternative management strategies for obstructive jaundice.
Response: We acknowledge the growing importance of EUS-guided interventions in the management of biliary obstructions and have incorporated findings from PMID: 39768654 into the Discussion section. Specifically, we now elaborate on EUS-guided biliary drainage (EUS-BD) as a minimally invasive alternative in cases where ERCP is unsuccessful or surgical intervention is not feasible. Furthermore, we have supplemented the discussion with details on percutaneous and endoscopic treatment options, enhancing the comprehensiveness of our manuscript.
- Annotated Figures for Imaging and Pathology
Reviewer’s Comment: The manuscript provides a good description of imaging and pathological findings. However, adding annotated figures for histological slides and imaging results (e.g., marking specific features like cystic wall thickening or ovarian-like stroma) would improve clarity for readers.
Response: Following your suggestion, we have updated Figure 2 to include additional annotations for histological and imaging findings. Specifically, we have added descriptions highlighting cystic wall thickening, improving the clarity and interpretability of the histopathological images. This adjustment ensures better alignment between the text and visual representation, allowing readers to more easily identify key diagnostic features.
- Discussion on Minimally Invasive Alternatives to Surgery
Reviewer’s Comment: The manuscript highlights the importance of early surgical resection in preventing malignancy and recurrence. It would benefit from a brief discussion on the risks and benefits of less invasive alternatives, such as percutaneous or endoscopic approaches, when surgery is not feasible.
Response: We agree with the importance of discussing alternative treatment strategies for cases where surgical resection is not an option. To address this, we have expanded the Discussion section to include:
- EUS-BD and other minimally invasive drainage techniques, including their indications, advantages, and potential limitations.
- Percutaneous interventions, which may provide temporary relief but have higher recurrence risks.
- Liver transplantation, which, although rarely needed, may be considered in cases with extensive biliary involvement or recurrent disease.
These additions ensure a more balanced discussion of treatment options tailored to different clinical scenarios.
Minor Comments
- Clarity and Grammar Improvements
Reviewer’s Comment: The manuscript is mostly clear but contains minor grammatical errors. For example, “the MBO to be a serious threat” should be clarified.
Response: We have carefully reviewed and refined the language in the manuscript to enhance clarity. Specifically, we have revised ambiguous statements in the case description to improve readability and precision.
- Readability of Surgical Description
Reviewer’s Comment: For smoother readability, sentences such as “the cyst wall and surface hepatic tissues were resected” could be rewritten.
Response: As suggested, we have restructured and reworded the relevant sections in the Case Presentation to improve clarity. The revised version ensures smoother readability and better comprehension of the surgical procedure.
- Consistency in Terminology
Reviewer’s Comment: Ensure consistent use of terms, e.g., "MCN-L" instead of alternating between "MCN-L" and "hilar MCN."
Response: We have reviewed the manuscript thoroughly and standardized the terminology, consistently using "MCN-L" throughout the text to avoid any inconsistencies.
- Optimization of the Timeline Figure (Figure 3)
Reviewer’s Comment: The timeline figure (Figure 3) is helpful but could be simplified for better readability. Consider enlarging key labels and focusing on the most critical clinical events.
Response: Following your recommendation, we have revised Figure 3 by:
- Enlarging key labels for improved visibility.
- Refining the descriptions of clinical events to highlight only the most relevant milestones.
- Enhancing the layout to ensure optimal readability.
These modifications improve the clarity and accessibility of the timeline for readers.
Final Remarks
Once again, we are grateful for your insightful suggestions, which have significantly enhanced the clarity, coherence, and scientific rigor of our manuscript. We believe that these revisions comprehensively address all concerns raised, and we look forward to your further feedback.

Round 2
Reviewer 1 Report
Comments and Suggestions for Authors
The authors have addressed the comments properly, substantially improving the quality of the manuscript.
Reviewer 2 Report
Comments and Suggestions for Authors
The authors have answered all questions and the paper can now be considered for publication.